# Multi-Source Probing for Open-Domain Conversational Understanding

**Yuanxi Li[1], Hao Zhou[2], Jie Zhou[2], Minlie Huang[3,4]**

[1]University of Illinois at Urbana-Champaign, IL, USA
[2]Pattern Recognition Center, WeChat AI, Tencent Inc., China
[3]The CoAI group, Tsinghua University, Beijing, China
[4]Department of Computer Science and Technology, Tsinghua University, Beijing, China

[1]yuanxi3@illinois.edu,[2]{tuxzhou, withtomzhou}@tencent.com
[3]aihuang@tsinghua.edu.cn

## Abstract

Dialogue comprehension and generation are vital to the success of open-domain dialogue systems. Although pre-trained generative conversation models have made significant progress in generating fluent responses, people have difficulty judging whether they understand and efficiently model the contextual information of the conversation. In this study, we propose a **M**ulti-**S**ource **P**robing (**MSP**) method to probe the dialogue comprehension abilities of open-domain dialogue models. MSP aggregates features from multiple sources to accomplish diverse task goals and conducts downstream tasks in a generative manner that is consistent with dialogue model pre-training to leverage model capabilities. We conduct probing experiments on seven tasks that require various dialogue comprehension skills, based on the internal representations encoded by dialogue models. Experimental results show that open-domain dialogue models can encode semantic information in the intermediate hidden states, which facilitates dialogue comprehension tasks. Models of different scales and structures possess different conversational understanding capabilities. Our findings encourage a comprehensive evaluation and design of open-domain dialogue models.

## 1 Introduction

Conversational understanding and response generation are critical for the success of open-domain dialogue systems. Recently, pre-trained open-domain dialogue models, including DialoGPT (Zhang et al., 2020), BlenderBot (Roller et al., 2020), and Meena (Adiwardana et al., 2020), have achieved impressive progress in a wide range of conversational tasks. The pre-trained model has become a solid foundation for the downstream fine-tuning process, such as generating empathetic (Zhong et al., 2020) and persona-coherent (Wolf et al., 2019b) responses, delivering knowledge-grounded conversations (Zhao et al., 2020; Wu et al., 2021), and completing task goals (Wu et al., 2020; Peng et al.,

2020). While these generative dialogue models can produce fluent responses, they still have many limitations in conversational understanding (Saleh et al., 2020; Li et al., 2016), leading to irrelevant, repetitive, and generic responses (Li et al., 2017a; Welleck et al., 2019; Cho and Saito, 2021).

Research on conversational understanding can provide a holistic evaluation of dialogue models (Parthasarathi et al., 2020), contributing to the deployment and design of models. However, the analysis of open-domain dialogue models on conversational understanding remains a controversial topic (Dinan et al., 2020; Tao et al., 2018; Ji et al., 2022). Some work (Sankar et al., 2019; Saleh et al., 2020) demonstrates that dialogue models have difficulty in capturing the conversational dynamics in the dialog history and struggle with conversational understanding tasks such as question answering, contradiction inference, and topic determination. In contrast, Parthasarathi et al. (2020) affirms the conversational understanding of open-domain dialogue models and indicates recurrent dialogue models perform better than transformer-based models. These studies have limitations in probing methods and experimental settings (Ravichander et al., 2020), making them not applicable to present large-scale open-domain dialogue models.

In this work, we propose a **M**ulti-**S**ource **P**robing (**MSP**) method to examine the conversational understanding ability of open-domain dialogue models. Specifically, MSP conducts dialogue comprehension tasks in a generative manner, which is coherent with the pre-trained dialogue generation task to take full advantage of model capabilities. In addition, considering that different tasks require various information of the dialogue context, MSP aggregates features from multiple sources to accomplish diverse tasks. We propose a multi-source cross-attention mechanism to extract local features and adopt a late fusion module to incorporate global features. With the help of these components, MSP

has the capability to evaluate generative dialogue models more accurately and comprehensively.

To expose the validity and reliability of our method, we conduct comprehensive experiments to compare the conventional MLP-based probing approach with MSP. Furthermore, we also set up a series of ablation experiments to verify the necessity of the multiple-source attention mechanism and the late-fusion module.

In order to verify that our method also applies to even more massive scale models, we extend our MSP method to provide insight into larger pretrained dialogue models. We found that larger models have stronger extraction capability for the information inferred via the pre-trained encoder.

Our study reveals three critical findings:

- Different from the conclusion reached by the vanilla probing method, we find through MSP that encoder hidden states contain more information than original embeddings in pretrained dialogue models, as reflected by the higher accuracy obtained on our probing tasks.

- Generative dialogue models with a single decoder have a worse understanding of the conversation than encoder-decoder-based models, as the uni-directional attention mechanism only encodes partial context (content before each token) information for tokens, leading to asymmetric representations of dialogue history and current utterance.

- Dialogue models can capture the dialogue structure in conversational understanding. Larger dialogue models have a better understanding of conversational information and achieve higher accuracy on probing tasks.

## 2 Related Work

### 2.1 Open-domain Conversational Models

Recently, open-domain conversation systems have been largely advanced due to the increase of dialogue corpus and the development of large-scale pre-training (Devlin et al., 2018; Radford et al., 2019; Brown et al., 2020). Pre-trained open-domain dialogue models, such as DialoGPT (Zhang et al., 2020), BlenderBot (Roller et al., 2020) and Meena (Adiwardana et al., 2020), demonstrate outstanding conversation skills, including empathetic (Zhong et al., 2020) and persona-coherent (Wolf et al., 2019b) response generation, delivering knowledge-grounded conversa-

tions (Zhao et al., 2020; Wu et al., 2021) and completing task goals (Wu et al., 2020; Peng et al., 2020). These dialogue models are capable of fluent response generation, but they still have many conversational understanding limits (Saleh et al., 2020; Li et al., 2016) that result in irrelevant, repetitive, and generic responses (Serban et al., 2017; Li et al., 2017a; Welleck et al., 2019; Cho and Saito, 2021). Several studies (Sankar et al., 2019; Bao et al., 2020) point out that generative dialogue models don't always properly exploit the existing dialog history and they are yet unable to understand the context to provide coherent and engaging conversations. Some work (Komeili et al., 2022; Zhou et al., 2018) introduce external knowledge to enhance conversational understanding, which facilitates the generation of relevant and coherent responses.

### 2.2 Probing Method

With the growing demand for natural language understanding, the probing method has been widely employed in machine translation (Belinkov et al., 2017, 2018; Dalvi et al., 2017; Yawei and Fan, 2021) and knowledge attribution (Alishahi et al., 2017; Beloucif and Biemann, 2021) to assess the linguistic properties of sentence representations learned by models.

Although several studies have been proposed to probe the conversational understanding capability of open-domain dialogue models (Dinan et al., 2020; Tao et al., 2018; Ji et al., 2022), this research area is still controversial. According to certain research (Sankar et al., 2019; Saleh et al., 2020; Das et al., 2020), conversational comprehension tasks including question answering, contradiction inference, and subject determination pose challenges for dialogue models in terms of capturing the conversational dynamics in the dialog history. In contrast, Parthasarathi et al. (2020) validates conversational comprehension of open-domain dialogue models and demonstrates that recurrent dialogue models outperform transformer-based models. In addition, previous work (Saleh et al., 2020; Alt et al., 2020; Ravichander et al., 2020; Parthasarathi et al., 2020; Richardson et al., 2020) usually adopted a certain probing method to perform model-level analysis of dialogue systems, thus they lacked an exhaustive comparison of different probing methods. These studies have reached opposite conclusions for two major reasons. First, these methods usually adopt a shallow Multi-Layer Perceptron (MLP) as the clas-

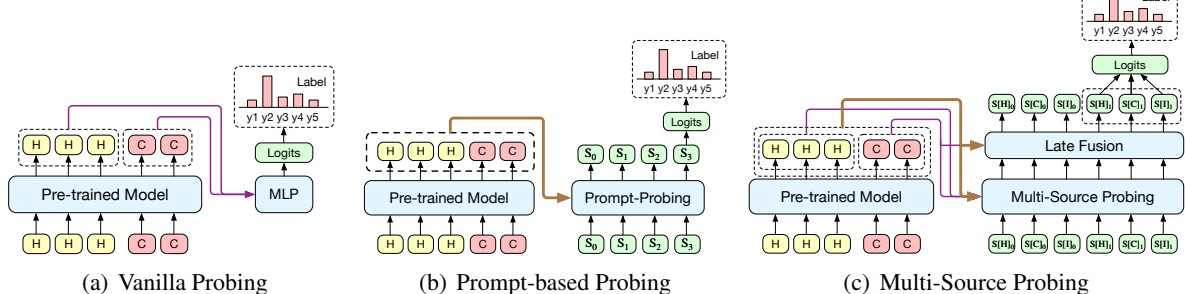

(a) Vanilla Probing      (b) Prompt-based Probing      (c) Multi-Source Probing

Figure 1: Overview of three probing methods. The MSP method introduces the multi-source attention mechanism to integrate local features from multiple sources and the late fusion module to capture global features, where probing tasks are conducted in a generative manner, which is consistent with the objective of the pre-training task.

sifier, which cannot fully utilize the information encoded in intermediate representations to conduct probing tasks. Second, the experimental settings are also insufficient (Ravichander et al., 2020), including the probing tasks and probed model scales.

## 3 Methodology

### 3.1 Vanilla Probing Approach

For the vanilla probing method, we first use pre-trained generative dialogue models to extract encoder hidden states and word embeddings corresponding to the probing task texts and then feed the extracted representations to a two-layer Multi-Layer Perceptron (MLP) classifier.

In this way, we calculate the accuracy of probing tasks, which is empirically assumed to reflect the ability of the corresponding model to capture information that is beneficial for the goal of dialogue understanding. Here only the parameters of the classifier are trainable during the training of probing tasks, with encoder parameters kept fixed. The vanilla probing method is shown in Fig 1(a).

First, we extract inner states from the encoder to get the word-level representations on **word embeddings** and **encoder states** in the entire probing task input $x = [u_1, \cdots, u_n]$, with $u_n = [w_1^{(n)}, \cdots, w_{c_n}^{(n)}]$ denoting the tokens in the $n_{th}$ utterance, where $c_n$ denotes the word numbers in the utterance $u_n$ and $\mathbf{w}_i^{(n)}$ is the word embedding of the $i_{th}$ word $w_i^{(n)}$ in the utterance $u_n$.

We require utterance-level representations for probing where mixed synthesized information is present since the probing tasks rely on high-level reasoning. To get $\mathbf{R}_{\text{history}}$ and $\mathbf{R}_{\text{current}}$, we independently averaged the representations corresponding to historical and contemporary utterances. Then

we concatenate the two averaged representations to get the final feature $\mathbf{R}_{\text{probing}}$ for probing tasks, where the concatenation operation is denoted by $\odot$. The process is defined as follows:

$$\mathbf{R}_{\text{history}} = \frac{1}{\sum_{t=1}^{n-1} c_t} \sum_{t=1}^{n-1} \mathbf{u}_t$$

$$= \frac{1}{\sum_{t=1}^{n-1} c_t} \sum_{t=1}^{n-1} \sum_{i=1}^{c_t} \mathbf{w}_i^{(t)}, \quad (1)$$

$$\mathbf{R}_{\text{current}} = \frac{1}{c_n} \sum_{i=1}^{c_n} \mathbf{w}_i^{(n)}, \quad (2)$$

$$\mathbf{R}_{\text{probing}} = \mathbf{R}_{\text{history}} \odot \mathbf{R}_{\text{current}}, \quad (3)$$

### 3.2 Multi-Source Probing Approach

The prior MLP-based approach has been considered fairly intuitive to detect if the dialogue model captures pertinent information in the encoder states. However, it is difficult to effectively utilize the information encoded by the dialogue models using these approaches due to the divergent objectives of downstream probing tasks and dialogue generation during pre-training (Liu et al., 2021; Schick and Schütze, 2021).

To address this issue, we propose a **M**ulti-**S**ource **P**robing (**MSP**) method to probe the dialogue comprehension abilities of open-domain dialogue models. MSP conducts probing tasks in a generative manner, which is consistent with the pre-trained task to take full advantage of model capabilities. As various probing tasks require information from different aspects of the dialogue, which may differ greatly from the dialogue generation task, we propose a multi-source attention mechanism to aggregate features from multiple sources to accomplish diverse tasks.

Moreover, the application of potential understanding capabilities that might be encoded in the decoder parameters is also lacking in MLP-based probing methods, while MSP reapplies the discarded information and provides better modeling of contextual information. The overview of MSP is presented in Figure 1(c).

### 3.2.1 Continuous Prompt Learning

Continuous (or soft) prompt learning is one of the central parts in our MSP method, which is shown in Figure 1(b), as the generative approach of probing classification is considered to be more consistent with the goal in the pretraining phase, allowing the model to be more adaptable.

Specifically, we set the Prompt-Template as a sequence of different soft tokens for decoder input. Each soft token has a unique word embedding that is constantly adjusted and updated during the training stage. Consistent with the vanilla probing setting, the transformer decoder is finetuned in MSP, while the encoder parameters are fixed.

Since our probing task is based on classification, a verbalizer (Hu et al., 2022) class is constructed here to project the original probability distribution over the whole vocabulary to the set of label words given by the probing task.

### 3.2.2 Multi-Source Attention

Different downstream probing tasks have various focuses on the location of the required information (Sankar et al., 2019), while the general attention module of dialogue models tends to fail in locating and extracting information from multiple aspects and sources in a fine-grained way.

Therefore, to avoid overshadowing the key information contained in the dialogue context when the significance of information is unevenly distributed, we propose a multi-source attention mechanism, by using multiple cross-attention masks corresponding to different sources through the decoding process, to generate more reasonable attention that can extract relevant local features from different sources for probing classification.

As shown in Figure 2, the multi-source attention module takes turns allocating different cross-attention to different parts of encoded representations for soft prompt tokens. We adopt three types of attention masks to operate the multi-source cross attentions, which are separately called **history-source**, **current-source**, and **integrated-source**. Only the relevant portion of the dialogue is given

attention by the history-source and current-source attention functions. While the integrated-source cross-attention mask is built to gather information from the full context of the dialogue.

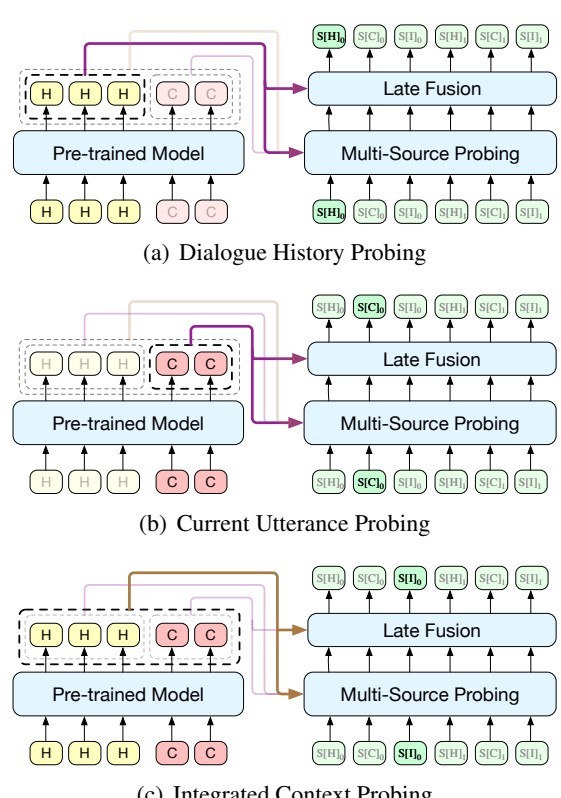

(a) Dialogue History Probing

(b) Current Utterance Probing

(c) Integrated Context Probing

Figure 2: Overview of multi-source attention.

In Prompt-Template, the three consecutive tokens are grouped into one combined unit, which is designed to perform a round of coverage during the probing process. We assign the three soft tokens in the last combined unit as the prediction position where the averaged logit is passed to the subsequent verbalizer function, and then we can calculate the loss for the corresponding classes. This step is specially formulated to fuse the information captured by various soft-token representations that are focused independently on the history, current, and entire section of the input text.

The decoder $f_\theta$ updates the hidden states $\mathbf{H}_i$ conditioned on the past decoder states $\mathbf{H}_{<i}$ with self-attention, the soft token embedding $\mathbf{S}_i$, as well as the encoded representations $\mathbf{E}_k$ with cross-attention, where $k$ could take values in $\mathcal{K} = \{\text{history}, \text{current}, \text{integrated}\}$, as follows:

$$\mathbf{H}_i = \begin{cases} f_\theta\left(\mathbf{H}_{<i}, \mathbf{S}_i, \mathbf{E}_{\text{his}}\right) & i = 0 \pmod 3 \\ f_\theta\left(\mathbf{H}_{<i}, \mathbf{S}_i, \mathbf{E}_{\text{cur}}\right) & i = 1 \pmod 3 \\ f_\theta\left(\mathbf{H}_{<i}, \mathbf{S}_i, \mathbf{E}_{\text{int}}\right) & i = 2 \pmod 3 \end{cases} \quad (4)$$

### 3.2.3 Late Fusion

Previous studies (Vig, 2019; Vig and Belinkov, 2019) have shown that the attention mechanism is sensitive to local features, while global features of conversations should also be considered to produce a comprehensive representation.

Thus we introduce the late fusion module as a merging strategy for information integration to generate a comprehensive representation from encoder hidden states, which is combined with the probing decoder states after passing through an MLP layer with dropout. Late fusion is designed to capture the global information encoded by the language model which serves as the role of a complement to the probing decoder aimed at extracting precise local features, thus allowing a higher sensitivity to various features of the linguistic information.

Here we implement the late fusion by averaging the encoder hidden states $\mathbf{E}_k$ that focus on the desired section of the text, where $k$ could take $\mathcal{K} = \{\text{history}, \text{current}, \text{integrated}\}$. We would receive the representation $\mathbf{L}_k$ after the late fusion, and $\mathbf{A}_k$ is the final representation obtained by combining the decoder hidden states $\mathbf{H}_k$ and the result from late fusion module.

$$\mathbf{L}_k = \mathbf{MLP}(\texttt{avg}(\mathbf{E}_k)), \quad (5)$$
$$\mathbf{A}_k = \mathbf{H}_k + \mathbf{L}_k. \quad (6)$$

The probability of $P_M(y|x)$ with probing label $y$ is calculated as:

$$P_M(y|x) = \texttt{softmax}(\mathbf{V}(\texttt{avg}(\{\mathbf{A}_k^{(n)}|k \in \mathcal{K}\}))). \quad (7)$$

where $\mathbf{V}$ is the language model head that projects the raw logits to the label space. Here $n$ is the number of combined units in the soft token sequence.

## 4 Experiment

### 4.1 Probing Tasks

**TREC** (Li and Roth, 2002) is a question classification dataset consisting of questions labeled with relevant answer types. The task aims to determine the information category the question is requesting.
**DialogueNLI** (Welleck et al., 2019) is a natural language inference task consisting of dialog turns with entailment, contradiction and neutral labels.
**MultiWOZ** (Eric et al., 2020) is a multi-domain, goal-directed conversational dataset for exploring natural language comprehension.

**Schema-Guided Dialog dataset (SGD)** (Rastogi et al., 2020) is an intent-tracking task that requires reasoning over multiple turns of dialogue.
**SNIPS** (Coucke et al., 2018) is an intent classification task with crowd-sourced, single-turn queries labeled for intent.
**ScenarioSA** (Zhang et al., 2019) is a sentiment classification task with turn-level sentiment labels and inputs from multi-turn, open-ended dialogues.
**DailyDialog Topic** uses dialogues from the DailyDialog dataset (Li et al., 2017b) to create a probing task where the goal is to make inferences about the topic of conversations (Saleh et al., 2020).

### 4.2 Probing Methods

**Vanilla MLP-based Probing** adopts a two-layer Multi-Layer Perceptron classifier (Saleh et al., 2020; Parthasarathi et al., 2020), which takes the word embedding and encoder states corresponding to the conversation context of the pre-trained dialogue models as input.
**Prompt-based Probing** applies prompt learning as the principle for performing probing tasks. During training, only the embeddings of soft prompt tokens and the verbalizer parameters are fine-tuned while the pre-trained encoder and decoder parameters are fixed (Liu et al., 2021).
**Multi-Source Probing** is the proposed approach, which is characteristic of its multi-source attention and late fusion module. The parameters of the verbalizer, prompt token embeddings, and decoders are fine-tuned during training.

### 4.3 Open-domain Dialogue Models

We first train three widespread generative dialogue models for 20 epochs on DailyDialog dataset (Li et al., 2017b), using the Maximum-likelihood objective(Sutskever et al., 2014). Specifically, we train Transformer from scratch and fine-tune BlenderBot-small and DialoGPT-small with pre-trained parameters on DailyDialog for a fair comparison, as adopted in prior work (Saleh et al., 2020).

Besides, we adopt BlenderBot-medium [400M] and DialoGPT-medium [345M] without fine-tuning for the inspection of larger pre-trained models.
**Transformer** (Vaswani et al., 2017) is a typical language model with multiple attention layers. We implement it in the form of encoder-decoder structure, with a 2-layer encoder and a 2-layer decoder.
**BlenderBot** (Roller et al., 2020) is a pre-trained dialogue model based on the encoder-decoder ar-

| Method | | TREC | DNLI | MWOZ | SGD | SNIPS | SSA | Topic |
|--------|------|------|------|------|------|------|------|-------|
| **Transformer** | | | | | | | | |
| **MLP** | Emb. | $83.1^*_{[1.07]}$ | $69.9_{[0.79]}$ | $92.1^*_{[0.38]}$ | $69.9^*_{[0.55]}$ | $97.9^*_{[0.24]}$ | $77.1_{[0.17]}$ | $53.3_{[2.00]}$ |
| | Enc. | $79.7_{[0.73]}$ | $68.7_{[0.70]}$ | $91.3_{[0.40]}$ | $67.0_{[0.66]}$ | $96.8_{[0.15]}$ | $77.8^*_{[0.36]}$ | $53.2_{[1.35]}$ |
| **PBP** | Emb. | $68.2_{[1.17]}$ | $52.0_{[0.66]}$ | $57.0^*_{[0.30]}$ | $53.7_{[0.56]}$ | $95.8^*_{[0.11]}$ | $64.6^*_{[0.25]}$ | $30.8_{[1.15]}$ |
| | Enc. | $69.1_{[1.14]}$ | $52.0_{[0.64]}$ | $56.5_{[0.26]}$ | $55.1^*_{[0.58]}$ | $94.9_{[0.56]}$ | $63.5_{[0.44]}$ | $35.1^*_{[1.31]}$ |
| **MSP** | Emb. | $90.5_{[0.27]}$ | $75.0_{[0.64]}$ | $95.2_{[0.27]}$ | $78.9_{[0.80]}$ | $97.9_{[0.11]}$ | $79.4_{[0.28]}$ | $60.7_{[1.19]}$ |
| | Enc. | $\mathbf{91.2}^*_{[0.50]}$ | $\mathbf{76.2}^*_{[0.23]}$ | $\mathbf{95.4}^*_{[0.21]}$ | $\mathbf{80.2}^*_{[0.48]}$ | $\mathbf{98.3}^*_{[0.21]}$ | $\mathbf{80.7}^*_{[0.29]}$ | $\mathbf{63.1}^*_{[1.61]}$ |
| **BlenderBot**SMALL | | | | | | | | |
| **MLP** | Emb. | $85.7_{[0.69]}$ | $72.4_{[0.23]}$ | $92.5_{[0.18]}$ | $72.9_{[0.26]}$ | $97.5_{[0.27]}$ | $78.7_{[0.37]}$ | $52.2_{[2.64]}$ |
| | Enc. | $92.1^*_{[0.56]}$ | $84.3^*_{[0.29]}$ | $93.6^*_{[0.32]}$ | $76.0^*_{[0.48]}$ | $97.7_{[0.18]}$ | $83.2^*_{[0.16]}$ | $65.2^*_{[0.78]}$ |
| **PBP** | Emb. | $75.5_{[2.18]}$ | $54.5_{[1.01]}$ | $58.9_{[0.67]}$ | $56.4_{[1.28]}$ | $95.6_{[0.64]}$ | $62.9_{[0.85]}$ | $50.8_{[2.12]}$ |
| | Enc. | $92.1^*_{[0.97]}$ | $61.0^*_{[3.30]}$ | $81.9^*_{[6.16]}$ | $80.6^*_{[2.74]}$ | $97.3^*_{[0.65]}$ | $76.3^*_{[0.76]}$ | $66.7^*_{[0.94]}$ |
| **MSP** | Emb. | $91.0_{[0.64]}$ | $77.9_{[1.00]}$ | $95.7_{[0.15]}$ | $82.0_{[1.34]}$ | $98.1_{[0.19]}$ | $80.3_{[0.26]}$ | $60.5_{[1.54]}$ |
| | Enc. | $\mathbf{93.8}^*_{[0.70]}$ | $\mathbf{87.4}^*_{[0.25]}$ | $\mathbf{96.5}^*_{[0.30]}$ | $\mathbf{88.8}^*_{[0.55]}$ | $\mathbf{98.6}^*_{[0.16]}$ | $\mathbf{84.4}^*_{[0.33]}$ | $\mathbf{71.3}^*_{[1.71]}$ |
| **DialoGPT**SMALL | | | | | | | | |
| **MLP** | Emb. | $88.4_{[0.63]}$ | $73.8_{[0.69]}$ | $92.9^*_{[0.29]}$ | $73.4^*_{[0.42]}$ | $98.8_{[0.14]}$ | $80.0_{[0.32]}$ | $52.6_{[1.22]}$ |
| | Enc. | $92.1^*_{[0.28]}$ | $80.9^*_{[0.30]}$ | $90.8_{[0.23]}$ | $72.8_{[0.24]}$ | $98.7_{[0.09]}$ | $81.1^*_{[0.23]}$ | $65.2^*_{[1.31]}$ |
| **PBP** | Emb. | $37.5_{[3.23]}$ | $42.3_{[1.27]}$ | $28.2_{[1.99]}$ | $10.6_{[0.52]}$ | $71.1_{[2.92]}$ | $57.1_{[0.01]}$ | $26.0_{[0.00]}$ |
| | Enc. | $52.0^*_{[1.17]}$ | $50.0^*_{[0.61]}$ | $44.9^*_{[0.60]}$ | $34.3^*_{[1.02]}$ | $86.2^*_{[0.62]}$ | $57.9^*_{[0.32]}$ | $36.7^*_{[1.12]}$ |
| **MSP** | Emb. | $94.2_{[0.38]}$ | $\mathbf{83.7}_{[0.39]}$ | $96.1_{[0.27]}$ | $85.4_{[0.03]}$ | $98.9_{[0.15]}$ | $82.3_{[0.46]}$ | $58.7_{[0.64]}$ |
| | Enc. | $\mathbf{96.8}^*_{[0.63]}$ | $83.4_{[0.21]}$ | $\mathbf{96.2}_{[0.62]}$ | $\mathbf{85.7}_{[0.43]}$ | $\mathbf{99.0}_{[0.64]}$ | $\mathbf{82.4}_{[0.37]}$ | $\mathbf{66.7}^*_{[0.66]}$ |

Table 1: Accuracy on probing tasks for **Transformer**, **BlenderBot**SMALL, **DialoGPT**SMALL. Experiments were conducted on three probing methods: 1) **MLP**: Vanilla MLP-based Probing 2) **PBP**: Prompt-based Probing. 3) **MSP**: Multi-Source Probing. Best results are marked in **bold**, and data that passed the significance test (t-test, p-value $< 0.05$) between word embeddings and encoder states under one specific probing method are super-scripted with an asterisk $^*$. The numbers in square brackets represent the standard deviation.

chitecture which is first pre-trained on 1.5B Reddit comment threads (Baumgartner et al., 2020) and later fine-tuned on Blended SkillTalk (BST) dataset (Smith et al., 2020).

**DialoGPT** (Zhang et al., 2020) is a dialogue response generation model for multi-turn conversations with a single decoder, which is pre-trained on large-scale Reddit data(Baumgartner et al., 2020).

## 5 Analysis

We will detail experimental results in this section, including the analysis of our main experiments on the performance of different probing methods, the ablation study of Multi-Source Probing architectures, and evaluations of different dialogue models and experimental settings.

### 5.1 Main Results

The main results are presented in Table 1, where each probing task is evaluated by calculating an average score of accuracy. We analyze the results from the following perspectives:

**Comparison between methods:** For all probing tasks and dialogue models, MSP achieves the best performances, indicating that our method more effectively leverages the relevant information encoded in the intermediate representations to conduct probing tasks. Besides, the majority of encoder state results outperform word embedding results, indicating that encoder states contain more semantic features than word embeddings. Thus, to some extent, dialogue models learn semantic information from conversations, which is required in conversational understanding tasks.

For the MLP-based probing method, we observe a similar phenomenon of the Transformer model that the performances of encoder states are not superior in many tasks as reported in prior work (Saleh et al., 2020). This observation demonstrates that the vanilla probing method has limitations in utilizing encoder states to conduct conversational understanding tasks, due to the gap between downstream classification tasks and the pre-trained dialogue generation task (Liu et al., 2021; Schick

| Method | | TREC | DNLI | MWOZ | SGD | SNIPS | SSA | Topic |
|---|---|---|---|---|---|---|---|---|
| **BlenderBot**$_{\text{MEDIUM}}$ | | | | | | | | |
| **MLP** | Emb. | $83.9_{[1.10]}$ | $73.3_{[0.77]}$ | $90.2_{[0.51]}$ | $65.3_{[0.80]}$ | $98.3_{[0.35]}$ | $79.3_{[0.28]}$ | $45.3_{[3.59]}$ |
| | Enc. | $91.1^*_{[0.20]}$ | $85.1^*_{[0.35]}$ | $90.6_{[0.36]}$ | $66.3_{[0.46]}$ | $98.1_{[0.12]}$ | $83.6^*_{[0.45]}$ | $56.2^*_{[1.20]}$ |
| **MSP** | Emb. | $92.6_{[0.27]}$ | $79.1_{[0.52]}$ | $96.3_{[0.31]}$ | $82.8_{[0.46]}$ | $98.1_{[0.66]}$ | $82.1_{[0.53]}$ | $61.5_{[0.51]}$ |
| | Enc. | $\mathbf{94.1}^*_{[0.44]}$ | $\mathbf{90.3}^*_{[0.23]}$ | $\mathbf{97.4}^*_{[0.65]}$ | $\mathbf{91.3}^*_{[0.48]}$ | $\mathbf{98.9}_{[0.61]}$ | $\mathbf{87.3}^*_{[0.50]}$ | $\mathbf{72.3}^*_{[1.09]}$ |
| **DialoGPT**$_{\text{MEDIUM}}$ | | | | | | | | |
| **MLP** | Emb. | $89.2_{[0.87]}$ | $74.5_{[0.55]}$ | $92.9_{[0.46]}$ | $74.6_{[0.58]}$ | $98.4_{[0.11]}$ | $80.4_{[0.35]}$ | $49.6_{[2.47]}$ |
| | Enc. | $94.8^*_{[0.13]}$ | $84.7^*_{[0.28]}$ | $91.5_{[0.20]}$ | $76.0_{[0.53]}$ | $98.6_{[0.12]}$ | $80.9_{[0.30]}$ | $67.0^*_{[0.94]}$ |
| **MSP** | Emb. | $95.2_{[0.53]}$ | $88.2_{[0.57]}$ | $\mathbf{96.5}_{[0.44]}$ | $86.1_{[0.49]}$ | $99.0_{[0.63]}$ | $82.8_{[0.53]}$ | $59.7_{[0.32]}$ |
| | Enc. | $\mathbf{97.4}^*_{[0.34]}$ | $\mathbf{89.5}^*_{[0.52]}$ | $96.4_{[0.63]}$ | $\mathbf{89.6}^*_{[0.38]}$ | $\mathbf{99.1}_{[0.42]}$ | $\mathbf{85.7}^*_{[0.80]}$ | $\mathbf{70.2}^*_{[0.92]}$ |

Table 2: The performance of MLP and MSP on two large-scale pre-trained dialogue models **BlenderBot**$_{\text{MEDIUM}}$ and **DialoGPT**$_{\text{MEDIUM}}$. Best results are marked in **bold**, and data that passed the significance test (t-test, p-value $< 0.05$) are super-scripted with an asterisk $^*$. The numbers in square brackets represent the standard deviation.

and Schütze, 2021). The prompt-based probing approach obtains the worst performance among the three probing methods. Although it performs the probing tasks in a generative manner, this approach cannot effectively extract relevant features required in conversational understanding tasks, leading to undesirable results.

**Comparison between models:** As the parameters of dialogue models increase, the performance on dialogue understanding tasks also improves based on the MSP method. However, DialoGPT doesn't outperform BlenderBot on some tasks and the performances of encoder states are not always significantly better than those of word embeddings. This is probably because DialoGPT adopts a single decoder structure with the uni-directional attention mechanism, which encodes partial context (content before each token) information for tokens, leading to asymmetric representations of dialogue history and current utterances. By contrast, the encoder-decoder-based BlenderBot applies the bi-directional attention mechanism to encode bi-directional information (content before and after each token) for tokens, achieving more consistent and superior performances on conversational understanding tasks.

We also evaluated the comprehension ability of pre-trained language models such as BERT, BART, and T5 through MSP, and the details are attached in Appendix 6.

## 5.2 Ablation Study

Here we set up a series of ablation experiments to investigate the validity and necessity of the components of our Multi-Source Probing approach, which is composed of two main parts: the multi-source

attention mechanism and the late fusion module. The multi-source attention mechanism is designed to pay fair attention to both the historical and current turns of the conversational context, while the design of the late fusion module is motivated by the fact that many probing tasks need to capture global features while maintaining a high sensitivity to local features, so we add a layer of late fusion after the decoder for fusing global and local features in the final representation. Due to the length limit, the results are presented in Table 4 in the Appendix.

### 5.2.1 Ablation Setting

Several sets of ablation experiments are designed to verify the necessity and effectiveness of different modules:

**MSP w/o LF** is an ablation setting where the late fusion module is removed compared to the complete MSP method.

**MSP w/o MS** is an ablation setting where the multi-source attention and late fusion module are removed. This approach has the same architecture as the prompt-based probing method except that the decoder is fine-tuned during training.

### 5.2.2 Effect of Late Fusion Module

In our experiment settings, we set ablation experiments on the effectiveness of late-fusion. Through the analysis, we discover that the late-fusion architecture enables a significant improvement in the ability of the probing model to extract the representational information encoded by the pre-trained language model. Furthermore, it not only incorporates the information provided by the encoder very well but also provides a nuanced insight for our probing architecture of representational informa-

| | Method | TREC | DNLI | MWOZ | SGD | SNIPS | SSA | Topic |
|---|---|---|---|---|---|---|---|---|
| | **Transformer** | | | | | | | |
| **MLP** | Rand.Emb. | $71.8_{[1.73]}$ | $60.4_{[0.42]}$ | $87.2_{[0.50]}$ | $60.5_{[0.60]}$ | $96.6_{[0.36]}$ | $70.6_{[0.29]}$ | $39.8_{[2.49]}$ |
| | Orig.Emb. | $83.1^{*}_{[1.07]}$ | $69.9^{*}_{[0.79]}$ | $92.1^{*}_{[0.38]}$ | $69.9^{*}_{[0.55]}$ | $97.9^{*}_{[0.24]}$ | $77.1^{*}_{[0.17]}$ | $53.3^{*}_{[2.00]}$ |
| | Shuf.Emb. | $83.0^{*}_{[1.08]}$ | $69.8^{*}_{[0.39]}$ | $92.0^{*}_{[0.46]}$ | $69.8^{*}_{[0.38]}$ | $97.9^{*}_{[0.35]}$ | $77.3_{[0.43]}$ | $53.5_{[1.44]}$ |
| | Shuf.Enc. | $77.2_{[0.66]}$ | $68.8_{[0.28]}$ | $90.1_{[0.27]}$ | $67.6_{[0.35]}$ | $96.6_{[0.36]}$ | $77.1_{[0.32]}$ | $55.9^{*}_{[2.17]}$ |
| **MSP** | Rand.Emb. | $80.4_{[0.72]}$ | $65.5_{[0.34]}$ | $94.2_{[0.41]}$ | $75.9_{[1.12]}$ | $97.5_{[0.21]}$ | $75.6_{[1.02]}$ | $37.2_{[1.76]}$ |
| | Orig.Emb. | $90.5^{*}_{[0.27]}$ | $75.0^{*}_{[0.64]}$ | $95.2^{*}_{[0.27]}$ | $78.9^{*}_{[0.80]}$ | $97.9^{*}_{[0.11]}$ | $79.4^{*}_{[0.28]}$ | $60.7^{*}_{[1.19]}$ |
| | Shuf.Emb. | $84.2_{[0.67]}$ | $68.9_{[0.41]}$ | $94.9_{[0.79]}$ | $76.9_{[0.67]}$ | $98.1_{[0.25]}$ | $77.7_{[1.43]}$ | $52.3_{[1.44]}$ |
| | Shuf.Enc. | $84.9_{[0.59]}$ | $70.0_{[0.24]}$ | $94.7_{[0.53]}$ | $78.1^{*}_{[0.23]}$ | $98.3_{[0.32]}$ | $78.8^{*}_{[0.95]}$ | $57.5^{*}_{[1.76]}$ |
| | **BlenderBot**SMALL | | | | | | | |
| **MLP** | Rand.Emb. | $76.2_{[1.10]}$ | $63.3_{[0.79]}$ | $90.6_{[0.48]}$ | $65.8_{[0.54]}$ | $97.2_{[0.29]}$ | $73.1_{[0.44]}$ | $40.1_{[2.12]}$ |
| | Orig.Emb. | $85.7^{*}_{[0.69]}$ | $72.4^{*}_{[0.23]}$ | $92.5^{*}_{[0.18]}$ | $72.9^{*}_{[0.26]}$ | $97.5_{[0.27]}$ | $78.7^{*}_{[0.37]}$ | $52.2^{*}_{[2.64]}$ |
| | Shuf.Emb. | $85.3^{*}_{[0.60]}$ | $72.7_{[0.51]}$ | $92.5^{*}_{[0.25]}$ | $72.9^{*}_{[0.28]}$ | $97.4_{[0.20]}$ | $78.8_{[0.43]}$ | $51.7_{[2.60]}$ |
| | Shuf.Enc. | $83.1_{[1.34]}$ | $79.8^{*}_{[0.22]}$ | $89.7_{[0.23]}$ | $68.0_{[0.44]}$ | $97.5_{[0.22]}$ | $78.5_{[0.20]}$ | $59.9^{*}_{[0.78]}$ |
| **MSP** | Rand.Emb. | $84.0_{[0.82]}$ | $74.3_{[0.25]}$ | $94.6_{[0.50]}$ | $82.2_{[0.53]}$ | $97.6_{[0.16]}$ | $76.4_{[0.75]}$ | $47.7_{[1.77]}$ |
| | Orig.Emb. | $91.0^{*}_{[0.64]}$ | $77.9^{*}_{[1.00]}$ | $95.7^{*}_{[0.15]}$ | $82.0_{[1.34]}$ | $98.1^{*}_{[0.19]}$ | $80.3^{*}_{[0.26]}$ | $60.5^{*}_{[1.54]}$ |
| | Shuf.Emb. | $88.7^{*}_{[0.66]}$ | $75.8_{[0.31]}$ | $95.8_{[0.29]}$ | $83.9_{[0.26]}$ | $98.1_{[0.09]}$ | $79.8_{[0.43]}$ | $56.3_{[1.50]}$ |
| | Shuf.Enc. | $85.2_{[0.45]}$ | $84.6^{*}_{[0.28]}$ | $95.4_{[0.46]}$ | $84.7^{*}_{[0.12]}$ | $98.3_{[0.15]}$ | $81.3^{*}_{[0.24]}$ | $65.9^{*}_{[4.16]}$ |
| | **DialoGPT**SMALL | | | | | | | |
| **MLP** | Rand.Emb. | $75.3_{[0.55]}$ | $74.1_{[0.61]}$ | $91.6_{[0.27]}$ | $68.2_{[0.48]}$ | $96.6_{[0.19]}$ | $75.3_{[0.44]}$ | $42.2_{[2.18]}$ |
| | Orig.Emb. | $88.4^{*}_{[0.63]}$ | $73.8_{[0.69]}$ | $92.9^{*}_{[0.29]}$ | $73.4^{*}_{[0.42]}$ | $98.8^{*}_{[0.14]}$ | $80.0^{*}_{[0.32]}$ | $52.6^{*}_{[1.22]}$ |
| | Shuf.Emb. | $89.5^{*}_{[0.84]}$ | $73.9_{[0.66]}$ | $93.0^{*}_{[0.31]}$ | $73.3^{*}_{[0.20]}$ | $98.9^{*}_{[0.17]}$ | $80.2^{*}_{[0.27]}$ | $50.8_{[1.62]}$ |
| | Shuf.Enc. | $80.1_{[0.59]}$ | $76.7^{*}_{[0.20]}$ | $86.9_{[0.58]}$ | $66.8_{[0.16]}$ | $96.7_{[0.24]}$ | $76.0_{[0.23]}$ | $65.4^{*}_{[0.32]}$ |
| **MSP** | Rand.Emb. | $87.8_{[0.32]}$ | $78.6_{[0.76]}$ | $92.3_{[0.26]}$ | $70.4_{[1.45]}$ | $97.1_{[0.12]}$ | $77.4_{[0.56]}$ | $52.2_{[0.87]}$ |
| | Orig.Emb. | $94.2^{*}_{[0.38]}$ | $83.7^{*}_{[0.39]}$ | $96.1^{*}_{[0.27]}$ | $85.4^{*}_{[0.03]}$ | $98.9^{*}_{[0.15]}$ | $82.3^{*}_{[0.46]}$ | $58.7^{*}_{[0.64]}$ |
| | Shuf.Emb. | $85.4_{[0.28]}$ | $81.2_{[0.24]}$ | $94.5_{[0.19]}$ | $81.5_{[1.02]}$ | $98.5_{[0.12]}$ | $80.3_{[0.39]}$ | $53.6_{[4.21]}$ |
| | Shuf.Enc. | $91.5^{*}_{[0.63]}$ | $82.7^{*}_{[0.26]}$ | $94.6_{[0.21]}$ | $82.3^{*}_{[0.34]}$ | $98.8^{*}_{[0.05]}$ | $81.5^{*}_{[0.21]}$ | $66.0_{[2.18]}$ |

Table 3: The performance with random embeddings and shuffled order of words in dialogue context. Data that passed the significance test (t-test, p-value $< 0.05$) are super-scripted with an asterisk $^{*}$.

tion at the shallow level. Consequently, this makes the structure more hierarchical and efficient in integrating different depths of encoded information.

We found from the results of our experiments that when the Multi-Source Probing Method does not have a late-fusion layer, the probing results of encoder states in several tasks are not better than those of word embeddings, which indicates that when we discard the late fusion, we also discard some classification local features needed for the task. We found that the ground-truth labels for these tasks are often determined by a combination of certain keywords in the historical utterance and some obvious prompt words in the current utterance of the dialog, so if we simply use the representations generated by the multi-layer transformer in the decoder, this tends to draw out only the global features and ignore the involvement of local features. So in this step of the experiment, we verified the significant role of the late fusion module for synthesizing local features.

### 5.2.3 Effect of Multi-Source Attention

We further developed ablation experiments to explore the impact of discarding the multi-source attention module of our MSP method, where an exciting discovery is found that the accuracy of the probing task was reduced by more than 10% when the MSP method did not efficiently utilize the multi-source attention module, suggesting that we need to introduce specialized designs to focus on different parts of the input text when evaluating the ability of a language model to understand a conversation. This demonstrates the need to introduce specialized designs to focus on different parts of the input text, which is a concern that generative dialogue models are constantly focusing on during the pre-training process. We experimentally found that a single cross-attention module is not effective in accomplishing our probing purpose, and thus

a multiple-source attention mechanism is a very central component of our probing approach.

## 5.3 Extension Experiments

Considering the undesirable performance of the prompt-based probing approach as shown in Table 1, we adopt MSP and MLP methods in the extension experiments.

### 5.3.1 Impact of Model Scale

Table 2 shows the performance of large-scale pre-trained conversational models on probing tasks. As can be seen, with the increase of model parameters, the performance on dialogue comprehension tasks also increases. The pre-trained dialogue models demonstrate a strong capability of conversational understanding even without fine-tuning on downstream dialogue corpus. For the MLP-based probing method, there is no obvious difference between the performance of encoder states and word embeddings in many tasks. By contrast, our approach is applicable to models of different scales from the 2-layer Transformer trained from scratch to the large-scale pre-trained BlenderBot and DialoGPT.

### 5.3.2 Impact of Word Embedding

During the training process of dialogue generation, word embeddings can learn and encode linguistic knowledge of conversations (Ravichander et al., 2020) as Word2Vec (Mikolov et al., 2013) and Glove (Pennington et al., 2014). Thus, we substitute the trained word embeddings with randomly initialized ones and conduct probing experiments to investigate the impact of word embeddings.

The performance with random embeddings is shown in Table 3. As we can see, there is a significant gap between the performance of random word embeddings and original ones, indicating that the trained word embeddings encode semantic information required by conversational understanding. In addition, the performance with random embeddings achieves above 90% on some tasks, such as SNIPS and MWOZ, while only obtaining less than 70% on others. It shows that different tasks require different degrees of conversational semantics.

### 5.3.3 Impact of Dialogue Structure

To examine whether dialogue models leverage dialogue structure in conversational understanding, we shuffle the order of the input tokens within dialogue history and the current utterance respectively. The results are shown in Table 3. We note that the MLP method weakens the feature of the word order by the average pooling operation, while MSP offers superior modeling of the word order.

We observed that the performances of both word embeddings and encoder states decrease substantially with shuffled input. Furthermore, the performance gap is even larger for encoder states, indicating that dialogue models can capture the context and flow of the dialogue for conversational understanding, rather than just processing individual words or utterances in isolation.

## 6 Conclusion

In this paper, we propose a Multi-Source Probing (MSP) method to probe the dialogue comprehension abilities of open-domain dialogue models. It conducts probing tasks in a generative manner that is consistent with the pre-training task of dialogue models. Besides, we propose the multi-source attention mechanism to aggregate features from multiple sources and the late fusion module to capture global features for downstream tasks. Our experimental results indicate the validity and reliability of the MSP method, which could also offer insight into the impact of the model scale, embedding quality, and dialogue structure on the conversational understanding capability of dialogue models when particular experimental settings are presented. This research underscores the importance of a comprehensive probing framework for dialogue models and paves the way for future studies aimed at enhancing their understanding capabilities.

## Limitations

Although the Multi-Source Probing (MSP) method can precisely detect the conversational understanding of open-domain dialogue models of different scales, it still faces two limitations. First, we focus on evaluating three widespread dialogue models in our experiments due to the limitation of computational resources. Dialogue models of different structures and scales could be probed with MSP in future work. Second, we adopt several representative classification tasks as our probing tasks, following previous work (Saleh et al., 2020). These tasks require different dialogue comprehension skills and have different degrees of difficulty, as analyzed in Section 5.3.2. In future work, a wide range of tasks of different complexity in different domains could be conducted based on MSP to construct a benchmark of conversational understanding.

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

| Method | | TREC | DNLI | MWOZ | SGD | SNIPS | SSA | Topic |
|---|---|---|---|---|---|---|---|---|
| **Transformer** | | | | | | | | |
| **MSP** | Emb. | $90.5_{[0.27]}$ | $75.0_{[0.64]}$ | $95.2_{[0.27]}$ | $78.9_{[0.80]}$ | $97.9_{[0.11]}$ | $79.4_{[0.28]}$ | $60.7_{[1.19]}$ |
| | Enc. | $\mathbf{91.2}^*_{[0.50]}$ | $\mathbf{76.2}^*_{[0.23]}$ | $\mathbf{95.4}^*_{[0.21]}$ | $\mathbf{80.2}^*_{[0.48]}$ | $\mathbf{98.3}^*_{[0.21]}$ | $\mathbf{80.7}^*_{[0.29]}$ | $\mathbf{63.1}^*_{[1.61]}$ |
| **MSP w/o LF** | Emb. | $90.0_{[0.54]}$ | $75.1_{[0.40]}$ | $95.2_{[0.19]}$ | $78.7_{[0.58]}$ | $97.9_{[0.42]}$ | $80.1^*_{[0.22]}$ | $61.6_{[2.17]}$ |
| | Enc. | $90.6_{[0.38]}$ | $75.4_{[1.05]}$ | $95.2_{[0.21]}$ | $79.7_{[0.54]}$ | $98.2_{[0.23]}$ | $79.4_{[0.42]}$ | $62.1_{[1.65]}$ |
| **MSP w/o MS** | Emb. | $87.7_{[0.30]}$ | $68.3_{[0.67]}$ | $84.4_{[0.35]}$ | $71.6^*_{[0.73]}$ | $97.7_{[0.41]}$ | $73.2_{[0.50]}$ | $60.5_{[1.65]}$ |
| | Enc. | $89.1^*_{[0.30]}$ | $69.2_{[0.93]}$ | $83.9_{[0.59]}$ | $70.2_{[0.75]}$ | $97.7_{[0.11]}$ | $72.8_{[0.30]}$ | $61.9_{[1.37]}$ |
| **BlenderBot**$_{\text{SMALL}}$ | | | | | | | | |
| **MSP** | Emb. | $91.0_{[0.64]}$ | $77.9_{[1.00]}$ | $95.7_{[0.15]}$ | $82.0_{[1.34]}$ | $98.1_{[0.19]}$ | $80.3_{[0.26]}$ | $60.5_{[1.54]}$ |
| | Enc. | $\mathbf{93.8}^*_{[0.70]}$ | $\mathbf{87.4}^*_{[0.25]}$ | $\mathbf{96.5}_{[0.30]}$ | $\mathbf{88.8}^*_{[0.55]}$ | $\mathbf{98.6}^*_{[0.16]}$ | $\mathbf{84.4}^*_{[0.33]}$ | $\mathbf{71.3}^*_{[1.71]}$ |
| **MSP w/o LF** | Emb. | $90.4_{[1.07]}$ | $74.7_{[0.30]}$ | $95.5_{[0.19]}$ | $81.9_{[0.67]}$ | $98.2_{[0.23]}$ | $79.8_{[0.47]}$ | $60.4_{[1.44]}$ |
| | Enc. | $93.6^*_{[0.62]}$ | $85.5^*_{[0.39]}$ | $96.4^*_{[0.32]}$ | $88.8^*_{[0.86]}$ | $98.6_{[0.31]}$ | $84.2^*_{[0.41]}$ | $71.2^*_{[0.27]}$ |
| **MSP w/o MS** | Emb. | $90.5_{[0.79]}$ | $71.7_{[0.59]}$ | $90.2_{[0.94]}$ | $81.2_{[0.17]}$ | $97.9_{[0.29]}$ | $74.0_{[0.56]}$ | $58.4_{[2.62]}$ |
| | Enc. | $93.4^*_{[0.25]}$ | $84.7^*_{[0.59]}$ | $95.8_{[0.21]}$ | $87.9^*_{[0.30]}$ | $98.5^*_{[0.06]}$ | $83.1^*_{[0.17]}$ | $70.7^*_{[2.95]}$ |
| **DialoGPT**$_{\text{SMALL}}$ | | | | | | | | |
| **MSP** | Emb. | $94.2_{[0.38]}$ | $\mathbf{83.7}_{[0.39]}$ | $96.1_{[0.27]}$ | $85.4_{[0.03]}$ | $98.9_{[0.15]}$ | $82.3_{[0.46]}$ | $58.7_{[0.64]}$ |
| | Enc. | $\mathbf{96.8}^*_{[0.63]}$ | $83.4_{[0.21]}$ | $\mathbf{96.2}_{[0.62]}$ | $\mathbf{85.7}_{[0.43]}$ | $\mathbf{99.0}_{[0.64]}$ | $\mathbf{82.4}^*_{[0.37]}$ | $\mathbf{66.7}^*_{[0.66]}$ |
| **MSP w/o LF** | Emb. | $93.0_{[0.69]}$ | $83.2^*_{[0.87]}$ | $95.7_{[0.54]}$ | $84.8^*_{[0.22]}$ | $98.7_{[0.25]}$ | $81.8_{[0.31]}$ | $58.3_{[2.75]}$ |
| | Enc. | $96.2^*_{[0.43]}$ | $81.7_{[0.73]}$ | $94.7_{[0.30]}$ | $79.1_{[0.60]}$ | $98.7_{[0.14]}$ | $81.6_{[0.20]}$ | $66.2^*_{[0.54]}$ |
| **MSP w/o MS** | Emb. | $93.4_{[0.52]}$ | $82.1_{[1.00]}$ | $92.6^*_{[0.52]}$ | $83.7_{[0.72]}$ | $98.7_{[0.19]}$ | $77.1_{[0.38]}$ | $56.3_{[1.72]}$ |
| | Enc. | $94.4_{[0.86]}$ | $83.0_{[0.41]}$ | $91.2_{[0.81]}$ | $77.5_{[1.27]}$ | $98.7_{[0.27]}$ | $80.7^*_{[0.27]}$ | $66.5^*_{[2.08]}$ |

Table 4: The performance of different dialogue models on probing tasks for ablation experiments. Here we introduce three ablation settings: 1) **MSP:** The complete Multi-Source Probing method, 2) **MSP w/o LF:** MSP without the late fusion module, 3) **MSP w/o MS:** MSP without the multi-source attention mechanism and the late fusion module. Best results are marked in **bold**, and data that passed the significance test ( t-test, p-value $< 0.05$) are super-scripted with an asterisk $^*$.

| Method | | TREC | DNLI | MWOZ | SGD | SNIPS | SSA | Topic |
|---|---|---|---|---|---|---|---|---|
| **MSP** | Emb. | $90.5_{[0.27]}$ | $75.0_{[0.64]}$ | $95.2_{[0.27]}$ | $78.9_{[0.80]}$ | $97.9_{[0.11]}$ | $79.4_{[0.28]}$ | $60.7_{[1.19]}$ |
| | Enc. | $\mathbf{91.2}^*_{[0.50]}$ | $\mathbf{76.2}^*_{[0.23]}$ | $\mathbf{95.4}^*_{[0.21]}$ | $\mathbf{80.2}^*_{[0.48]}$ | $\mathbf{98.3}^*_{[0.21]}$ | $\mathbf{80.7}^*_{[0.29]}$ | $\mathbf{63.1}^*_{[1.61]}$ |
| **MLP-Deep** | Embs. | $83.7^*_{[1.70]}$ | $70.2_{[0.48]}$ | $92.0^*_{[0.36]}$ | $69.2^*_{[0.67]}$ | $97.7^*_{[0.20]}$ | $77.3_{[0.39]}$ | $56.4_{[1.10]}$ |
| | Enc. | $80.7_{[1.39]}$ | $69.3_{[0.62]}$ | $89.5_{[0.74]}$ | $67.5_{[0.67]}$ | $96.9_{[0.30]}$ | $78.4^*_{[0.24]}$ | $56.8_{[2.72]}$ |

Table 5: The performance of the **MSP** and **MLP-Deep** methods on probing tasks. Best results are marked in **bold**, and data that passed the significance test ( t-test, p-value $< 0.05$) are super-scripted with an asterisk $^*$.

# A Discussion

## A.1 Eliminate the Effect of Parameter Scales

We have already found a positive correlation between parametric size and probe model performance earlier, so here we further explore the gap between MLP and MSP performance at the same parameter size to address the concern about the effect of the number of parameters on the probing results.

We extended the original two-layer MLP to a parameter scale consistent with the MSP, and the results are attached in Table 5. Although the parameter size of the MLP is increased to a level comparable to that of the MSP, the results are still sub-optimal because it does not operate in a manner consistent with the goals in the pretraining phase of the dialogue model. The supplementary experiment results demonstrate the effectiveness of the MSP structure.

## A.2 Generalization on Pre-trained Language Models

We also evaluated the results of MSP over the state-of-the-art pre-trained language models. According to the results over BERT, BART, and T5 in Table 6, we could conclude that MSP still outperforms MLP even on the pre-trained language model.

Among the three models of comparable parameter size, Bart has the most outstanding ability for dialogue understanding. Bert performs optimally on the topic classification task, and T5 has very good performance on the task of intent detection, which is consistent with the characteristics and pre-training goals of the individual models themselves.

We found that Blenderbot-Small outperformed all of the three general pre-trained models in terms

| Method | | TREC | DNLI | MWOZ | SGD | SNIPS | SSA | Topic |
|---|---|---|---|---|---|---|---|---|
| **Bert MLP** | Embs. | $87.8^*_{[0.55]}$ | $73.7_{[0.77]}$ | $92.9^*_{[0.44]}$ | $73.0^*_{[0.68]}$ | $98.4^*_{[0.19]}$ | $78.2_{[0.34]}$ | $46.9_{[1.88]}$ |
| | Enc. | $85.7_{[0.24]}$ | $81.8^*_{[0.45]}$ | $90.2_{[0.39]}$ | $70.2_{[0.34]}$ | $95.8_{[0.19]}$ | $81.1^*_{[0.28]}$ | $59.9^*_{[1.08]}$ |
| **Bert MSP** | Embs. | $92.7_{[0.36]}$ | $80.2_{[0.54]}$ | $95.4_{[0.18]}$ | $81.9_{[0.37]}$ | $98.5_{[0.56]}$ | $81.9_{[0.47]}$ | $58.4_{[1.54]}$ |
| | Enc. | $\mathbf{94.9}^*_{[0.23]}$ | $\mathbf{86.7}^*_{[0.59]}$ | $\mathbf{95.8}_{[0.42]}$ | $83.2^*_{[0.44]}$ | $98.6_{[0.68]}$ | $82.3_{[0.97]}$ | $\mathbf{69.7}^*_{[1.12]}$ |
| **Bart MLP** | Embs. | $87.5_{[0.57]}$ | $75.1_{[0.95]}$ | $93.2_{[0.50]}$ | $73.7_{[0.60]}$ | $98.5_{[0.12]}$ | $79.9_{[0.12]}$ | $48.2_{[2.13]}$ |
| | Enc. | $89.8_{[0.29]}$ | $85.9^*_{[0.36]}$ | $91.9_{[0.24]}$ | $72.5_{[0.39]}$ | $98.8_{[0.05]}$ | $83.7^*_{[0.14]}$ | $56.0^*_{[0.75]}$ |
| **Bart MSP** | Embs. | $91.3_{[0.33]}$ | $81.3_{[0.87]}$ | $96.1_{[0.25]}$ | $84.5_{[0.79]}$ | $98.4_{[0.23]}$ | $81.7_{[0.21]}$ | $60.5_{[1.74]}$ |
| | Enc. | $\mathbf{95.6}^*_{[0.29]}$ | $\mathbf{87.6}^*_{[0.46]}$ | $\mathbf{96.3}_{[0.41]}$ | $\mathbf{87.2}^*_{[0.07]}$ | $\mathbf{99.0}^*_{[0.18]}$ | $\mathbf{84.9}^*_{[0.22]}$ | $68.1^*_{[0.76]}$ |
| **T5 MLP** | Embs. | $86.8_{[1.54]}$ | $74.0_{[1.07]}$ | $93.1_{[0.54]}$ | $73.6^*_{[0.16]}$ | $97.2_{[0.38]}$ | $78.5_{[0.86]}$ | $44.5_{[5.33]}$ |
| | Enc. | $89.5^*_{[0.26]}$ | $83.3^*_{[0.34]}$ | $93.1_{[0.28]}$ | $72.4_{[1.22]}$ | $98.1_{[0.11]}$ | $82.1^*_{[0.20]}$ | $57.9^*_{[2.81]}$ |
| **T5 MSP** | Embs. | $91.7_{[0.41]}$ | $80.8_{[1.39]}$ | $95.6_{[0.47]}$ | $83.2_{[0.19]}$ | $98.6_{[0.07]}$ | $81.4_{[0.24]}$ | $59.1_{[2.45]}$ |
| | Enc. | $\mathbf{93.2}^*_{[0.73]}$ | $\mathbf{85.1}^*_{[0.54]}$ | $\mathbf{96.2}_{[0.58]}$ | $\mathbf{85.7}^*_{[0.02]}$ | $\mathbf{98.9}_{[0.17]}$ | $\mathbf{83.3}^*_{[0.25]}$ | $\mathbf{67.6}^*_{[1.22]}$ |

Table 6: The performance of the MSP and MLP methods with the state-of-the-art pre-trained language models **BERT**, **BART**, and **T5** on several probing tasks. Best results are marked in **bold**, and data that passed the significance test ( t-test, p-value $< 0.05$) are super-scripted with an asterisk $^*$.

| Method | | TREC | DNLI | MWOZ | SGD | SNIPS | SSA | Topic |
|---|---|---|---|---|---|---|---|---|
| **MLP** | Rand.Emb. | $71.8_{[1.73]}$ | $60.4_{[0.42]}$ | $87.2_{[0.50]}$ | $60.5_{[0.60]}$ | $96.6_{[0.36]}$ | $70.6_{[0.29]}$ | $39.8_{[2.49]}$ |
| | Orig.Emb. | $83.1^*_{[1.07]}$ | $69.9^*_{[0.79]}$ | $92.1^*_{[0.38]}$ | $69.9^*_{[0.55]}$ | $97.9^*_{[0.24]}$ | $77.1^*_{[0.17]}$ | $53.3^*_{[2.00]}$ |
| **Linear** | Rand.Emb. | $65.7_{[1.70]}$ | $53.4^*_{[0.59]}$ | $84.9_{[0.30]}$ | $55.4_{[0.26]}$ | $95.5_{[0.46]}$ | $69.2^*_{[0.25]}$ | $35.8^*_{[2.30]}$ |
| | Orig.Emb. | $66.0_{[0.74]}$ | $51.2_{[0.85]}$ | $84.2_{[0.61]}$ | $63.8^*_{[0.98]}$ | $95.5_{[0.38]}$ | $62.3_{[0.61]}$ | $31.2_{[3.26]}$ |

Table 7: The performance of the **MLP** and **Linear-Probing** methods with original and random embeddings on several different probing tasks. Data that passed the significance test (t-test, p-value $< 0.05$) are super-scripted with an asterisk $^*$.

| Model | | Parameter | Perplexity |
|---|---|---|---|
| **Transformer** | | 37M | 30.3 |
| **BlenderBot**SMALL | | 90M | 10.4 |
| **DialoGPT**SMALL | | 117M | 8.6 |

Table 8: Training results of dialogue models.

of accuracy on the MWOZ SGD and Topic tasks. While DialoGPT-Small performs best on the TREC and SNIPS tasks. Another point to note is that all three models included in the supplemental experiments have over 25% more parameters than the corresponding BlenderBot-Small DialoGPT.

### A.3 Non-linear Probing Finetuning

Nonlinear probing is widely adopted in previous works (Belinkov et al., 2017; Belinkov and Glass, 2017; Conneau et al., 2018). In fact, non-linear probing and linear probing are essentially similar in probing. Classifiers with even a shallow linear structure can still fit well on these probing tasks. To this end, we add an experiment on linear probing, in which the original/random embeddings of the dialogue model were connected to a single linear layer for linear regression. The results in Table 7 show

that the linear probing method can also achieve more than 95% accuracy on random embeddings in the SNIPS task. It also proves the previous conclusion that linear probing is not an exclusive probing skill and non-linear probing has stronger probing performance in many aspects. We also add the experimental results of MSP on Bert (See Table 6), where we could see that Bert does not outperform pretrained ODD models of comparable size on the task of probing for language understanding.

## B Dataset Examples

Examples of probing dataset are shown in Table 9.

## C Implementation Details

We implemented the above models with PyTorch (Paszke et al., 2017), OpenPrompt (Ding et al., 2021) and Huggingface Library (Wolf et al., 2019a). When implementing the Multi-Source Probing method upon DialoGPT, we introduced randomly initialized cross-attention parameters together with the decoder for fine-tuning since DialoGPT is not an encoder-decoder-based model. We utilized Soft Verbalizer (Hambardzumyan et al., 2021; Hu et al., 2022), where a continuous vector

| Dataset | Type | Example | Class | Label |
|---------|------|---------|-------|-------|
| TREC | Question Classification | [User1]: What sprawling U.S. state boasts the most airports? | number, entity, location, ... | location |
| DialogueNLI | Natural Language Inference | [User1]: My locks are chestnut .
[User1]: You are blonde . | entail, contradict, neutral | contradict |
| MultiWOZ | Intent Classification | [User1]: Can you find an expensive restaurant that serves Venetian food?
[User2]: Sorry, looks like there aren't any Venetian restaurants that are expensive. Would you like something else?
[User1]: Yes, I would be interested in one that serves Chinese food. Where would you recommend? | restaurant-inform, taxi-request, general-thank, ... | restaurant-inform |
| Schema-Guided | Intent Classification | [User1]: Can you show me attractions I can visit?
[User2]: Where do you want me to search.
[User1]: In Toronto please.
[User2]: I found 10, how about 3D Toronto Sign, it's a tourist attraction. | find-attractions, reserve-flight, get-ride, ... | find-attractions |
| SNIPS | Intent Classification | [User1]: I need another artist in the New Indie Mix playlist. | search-screening, add-to-playlist, get-weather, ... | add-to-playlist |
| ScenrioSA | Sentiment Classification | [User1]: Have you met the new intern?
[User2]: Yes. She's very enthusiastic.
[User1]: I know. I don't trust her.
[User2]: Why? She is just ambitious.
[User1]: I think she wants my job. | positive, negative, neutral | negative |
| DailyDialog Topic | Topic Classification | [User1]: Happy birthday , Jim ! Here is a present for you.
[User2]: Oh , great ! I love it!
[User1]: I'm very glad to hear that .
[User2]: Come here, let me introduce some friends to you. | ordinary life, work, school, tourism, politics, relationship, ... | relationship |

Table 9: Examples from probing tasks.

is designed for each class label, to generate the probability distribution for class label space by calculating the dot product between the output of the language model and the class vector. The class vectors are initialized with the pretrained token embeddings and will be fine-tuned through training.

All models in this paper are optimized through ADAM (Kingma and Ba, 2014) with learning rate and dropout rate optimized through grid search. The number of soft tokens is empirically set to 12 in our Multi-Source Probing method. The learning rate is searched within the range of $\{5 \times 10^{-5}, 1 \times 10^{-4}, 2 \times 10^{-4}, 3 \times 10^{-4}, 5 \times 10^{-4}\}$ and dropout rate within $\{0.1, 0.2, 0.3, 0.4, 0.5\}$. The batch size for Transformer, BlenderBot, and DialoGPT are 128, 64, and 16 respectively. Accuracy and standard deviation data in this paper are calculated from the results of 5 replicate experiments. We conducted our probing experiments on the NVIDIA V100 Tensor Cores, the average run-time for each probing task is about 5 hours.