# OpenReview forum: "Multi-Source Probing for Open-Domain Conversational Understanding"
_EMNLP/2023/Conference — EMNLP 2023 Main_

### Official Review · Reviewer_i7vA · 2023-08-03

**Soundness:** 4

**Excitement:**

2: Mediocre: This paper makes marginal contributions (vs non-contemporaneous work), so I would rather not see it in the conference.

**Paper Topic And Main Contributions:**

This paper introduces a Multi-Source Probing (MSP) method to evaluate the dialogue comprehension abilities of open-domain dialogue models. The authors conduct probing experiments on seven tasks based on the internal representations encoded by dialogue models. The experimental results show that open-domain dialogue models can encode  semantic information in their intermediate hidden states, and different models have different conversational understanding capabilities.  The findings encourage a comprehensive evaluation and design of open-domain dialogue models.

**Questions For The Authors:**

1. The model's performance has already achieved excellent results in certain tasks. In light of this, does probing for this particular task still hold sufficient significance?

**Reasons To Accept:**

1. The writing in the paper is clear and  easy to understand
2. Extensive experiments were conducted across multiple tasks and settings.

**Reasons To Reject:**

1. I have some concerns about the necessity of employing complex modeling for the probing task. If a significant amount of human intuition and experience is incorporated to enhance the model's performance, can we still consider the test results as indicative of the probing model's inherent capabilities?
2. After reading the entire paper, it is not clear what new experimental phenomena, insights, or differences from prior work have been derived from this study. It would be beneficial to highlight any novel contributions or insights that could inspire further research in the field.

**Reproducibility:**

5: Could easily reproduce the results.

**Reviewer Confidence:**

2: Willing to defend my evaluation, but it is fairly likely that I missed some details, didn't understand some central points, or can't be sure about the novelty of the work.

---

> ### Author Rebuttal · Authors · 2023-08-29
>
> Thank you for your time!
>
> **Q1: Concerns about whether incorporating human intuition into complex probing models skews the test results.**
>
> First, it's important to clarify that the objective of the MSP technique is to provide a comprehensive and nuanced evaluation of dialogue models. While it may seem complex, the intention behind using multiple data sources and generative tasks is to simulate real-world, multifaceted scenarios more effectively than simpler probing methods.
>
> Any model or evaluation technique is ultimately based on human-generated benchmarks or intuitions. The goal here is not to entirely eliminate human intuition but to systematically operationalize it. MSP offers a structured approach to incorporate what we consider essential tasks for dialogue comprehension, which aims for a more generalized evaluation as it encompasses multiple sources and tasks. This helps in offsetting the specific biases or limitations that might be ingrained due to human intuition, making it more transparent for the probing to assess the capabilities of the model.
>
> ---
>
> **Q2: Unclear novel contributions in the paper; suggests highlighting unique insights for future research.**
>
> Firstly, our Multi-Source Probing (MSP) method represents a significant departure from existing approaches in the open-domain dialogue landscape. Unlike previous MLP-based probing methods, MSP effectively addresses the misalignment between the objectives of downstream probing tasks and the goals of dialogue generation during the pretraining phase. This ensures a more harmonized evaluation of dialogue systems(L230-235).
>
> Secondly, our study is pioneering in its introduction of a prompt-based design into probing methodologies. MSP conducts probing tasks in a generative manner, which is better aligned with the original pretraining tasks of dialogue models. This synchronization is a step forward in the dialogue probing field, offering a more cohesive evaluation approach(L239-254).
>
> In terms of empirical evidence, our paper presents results across a range of experimental settings (as detailed in Tables 1, 2, and 3). These results corroborate the effectiveness of MSP in providing a more nuanced evaluation for pre-trained dialogue models. The comparative analysis between MSP and traditional MLP-based methods (Table 1) highlights the advantages of our generative probing approach. Furthermore, the ablation studies underscore the importance of innovative MSP modules like late fusion and multi-source attention (Table 4).
>
> Our study reveals three critical insights:
> 1. Different from the conclusion reached by vanilla probing method, we find through MSP that encoder hidden states contain more information than original embeddings in pretrained dialogue models, as reflected by the higher accuracy obtained on our probing tasks.
> 2. Generative dialogue models with a single decoder have a worse understanding of the conversation than encoder-decoder-based models, as the uni-directional attention mechanism only encodes partial context (content before each token) information for tokens, leading to asymmetric representations of dialogue history and current utterance.
> 3. Dialogue models can capture the dialogue structure in conversational understanding. Larger dialogue models have a better understanding of conversational information and achieve higher accuracy on probing tasks.
>
> In summary, both the architectural design and empirical validation of MSP attest to its innovative nature, offering new avenues for future research in dialogue system evaluation.
>
> ---
>
> **Q3: Questioning the necessity of probing given the model's already excellent performance in certain tasks**
>
> Thank you for raising an insightful question that speaks to the core considerations behind our probing methodology.
>
> While the model has indeed demonstrated high performance in certain tasks, this doesn't eliminate the necessity for a more nuanced evaluation through probing. Probing serves a dual purpose: not only to assess task-specific performance but also to scrutinize the model's ability to discern between varying levels of informative features. Our probing technique, Multi-Source Probing (MSP), accentuates these differences by providing a more detailed examination of the model's true capabilities and shortcomings in understanding dialogue.
>
> Moreover, our work aims to discern the underlying factors contributing to the model's performance. It's essential to distinguish whether the model's excellence is attributed to its nuanced understanding of the underlying features and informational structures in dialogues, as opposed to mere sensitivity to dataset-specific shortcuts, such as individual vocabulary items or syntactic structures. In other words, we are keen on ascertaining that the model's prowess is rooted in genuine comprehension rather than trivial memorization or exploitation of data-specific cues.
>
> Therefore, probing continues to hold significant value as a comprehensive evaluation tool, offering insights into the model's capabilities that go beyond mere task performance metrics.

---

### Official Review · Reviewer_vySG · 2023-08-04

**Soundness:** 3

**Excitement:**

4: Strong: This paper deepens the understanding of some phenomenon or lowers the barriers to an existing research direction.

**Paper Topic And Main Contributions:**

The paper is about determining what is the best way to measure the dialog comprehension ability of open-domain dialog models. There are several methods proposed which involve leveraging the word embeddings along with encoder states to perform a variety of NLP tasks (question answering, nli etc.)

The main contribution of this work is the proposal of their novel method.

**Reasons To Accept:**

(1) The experimental results look convincing that their method is better than previous probing methods based on their selection of datasets and models.

**Reasons To Reject:**

(1) The description of their MSP and Prompt Based methods are not super clear. It is not defined what are these soft tokens and prompt templates.

(2) It is mentioned that decoder models are worse than encoder-decoder models. If this is the case how does this affect the actual use of these models when interacting with humans. Do those results align?

(3) There was a mention that one controversy in this problem space is that recurrent methods are better than transformer models. If this is the case then should these experiments be run on recurrent models too?

**Reproducibility:**

4: Could mostly reproduce the results, but there may be some variation because of sample variance or minor variations in their interpretation of the protocol or method.

**Reviewer Confidence:**

4: Quite sure. I tried to check the important points carefully. It's unlikely, though conceivable, that I missed something that should affect my ratings.

**Typos Grammar Style And Presentation Improvements:**

I would try to put the ablation results in the main text.

---

> ### Author Rebuttal · Authors · 2023-08-29
>
> Thank you for your valuable comments!
>
> **Q1. Definition of soft tokens and prompt templates in MSP and PBP**
>
> Soft (Continuous) prompt tokens generally refer to tunable, learnable tokens that are optimized during fine-tuning. Unlike hard tokens such as fixed words, these soft tokens can adapt during the training process to help guide the model toward better performance on specific tasks.
>
> As for prompt templates, they are essentially predefined scaffolds for creating prompts. In this context, a well-designed prompt template can help extract more accurate and nuanced answers from the model. Here our prompt template is a sequence of soft tokens for decoder input, and each soft token in this sequence has a unique word embedding that is constantly adjusted and updated during the training stage.
>
> By incorporating these concepts of soft tokens and prompt templates, both our MSP and Prompt-Based methods aim to offer a more effective way of probing dialog models, providing richer insights into their understanding and performance capabilities.
>
> ---
>
> **Q2: Question on the impact of decoder models' inferior performance on human interaction and result alignment**
>
> Thank you for asking a critical question that delves into the practical implications of our findings. Our research highlights the superiority of encoder-decoder models like BlenderBot over decoder-only models like DialoGPT in capturing the intricacies of conversational context. This suggests that encoder-decoder models, with their bi-directional attention mechanisms, are better equipped to generate richer and more context-aware dialogues in human interactions.
>
> However, the real-world application of these models in human-machine dialogue can be influenced by a multitude of factors, such as the characteristics of conversation, and the particular use case. Decoder-only models may still be highly effective in straightforward or less context-dependent scenarios, whereas encoder-decoder models would likely excel in complex, context-rich conversations.
>
> Additionally, as noted in our Extension Experiment section (Sect 5.3.1 and Table 2), the performance gap between decoder-only and encoder-decoder models narrows significantly as the model scales in terms of parameters. In other words, when dealing with large models like LAMA and GPT-4, which are decoder-only structures, their performance is comparably exceptional. This suggests that the sheer volume of parameters could have a more pronounced influence on performance than the specific architectural design.
>
> Therefore, while the architecture does have an impact, particularly for smaller models, its importance may be overshadowed by other factors, most notably the scale of the model, when evaluating efficacy in human interactions.
>
> ---
>
> **Q3: Question about running experiments on recurrent models due to their purported superiority over transformers**
>
> Thank you for raising the question about the potential relevance of RNNs in our research context. While the debate over RNNs vs. Transformers exists, there are compelling reasons why we focused on Transformer models for our experiments.
>
> 1. The transformer models we used, such as BlenderBot and DialoGPT, are currently the most advanced mainstream dialogue model architectures. They offer several advantages, including ease of training deep models and resistance to overfitting. In contrast, RNN models struggle to optimize well at large parameter scales, which means we couldn't find RNN models of comparable scale and performance to BlenderBot and DialoGPT for a fair experiment.
>
> 2. Our probing methods have been designed to significantly test the understanding capabilities of dialogue models. Therefore, the primary objective of our study was not to reignite the RNN vs. Transformer debate but to introduce a more effective probing methodology.  Our probing method has demonstrated a significant capacity to assess the understanding ability of dialogue models, thereby highlighting the limitations of previous methods and conclusions, regardless of whether those were based on RNN or Transformer architectures.
>
> 3. The paper we referred to[7] already explored experiments on RNNs and reached conclusions that were consistent with those of Transformer models. Given this and the space constraints for our own paper, it was logical for us to utilize Transformers as our baseline.
>
> In summary, while it would be interesting to run experiments on recurrent models, our decision to focus on transformer models was driven by their superior performance and scalability, the effectiveness of our probing methods, and the consistency of previous findings between RNNs and transformer models.
>
> ---
>
> **References:**
>
> [7]https://aclanthology.org/2020.nlp4convai-1.15/

---

### Official Review · Reviewer_PXn6 · 2023-08-10

**Soundness:** 3

**Excitement:**

3: Ambivalent: It has merits (e.g., it reports state-of-the-art results, the idea is nice), but there are key weaknesses (e.g., it describes incremental work), and it can significantly benefit from another round of revision. However, I won't object to accepting it if my co-reviewers champion it.

**Paper Topic And Main Contributions:**

This paper aims at examining the dialogue context modeling ability of open-domain dialogue models with probing approaches. To fully utilize the information encoded in the intermediate representations, the authors propose a Multi-Source Probing method, which takes advantage of multiple cross-attention masks to attend to different representations from the pre-trained dialogue encoder through the decoding process with soft prompt tokens, including the history-source, current-source, and integrated source. Then these local features are gathered through a late fusion layer for integrating global features. Finally, the output representations of three consecutive prompt tokens are averaged to get the final prediction with a verbalizer. Extensive experiments covering seven probing tasks, three probing methods, and five dialogue models show the following conclusions: dialogue models learn semantic information from conversations; decoder models are weaker than encoder-decoder models; the pre-trained dialogue models show a strong er capability when scaled up; the pre-trained word-embeddings are necessary; and the token order matters.

**Questions For The Authors:**

Question A: Table 1, which lines of data are "data that passed the significance test" compared with?

Question B: Is there a relationship between the performance of the probing method and the number of fine-tuned parameters of each method? Why is the decoder fixed for Prompt-based Probing while fine-tuned for Multi-Source Probing?

**Reasons To Accept:**

1. The paper is easy-to-follow and self-contained with the introduction of previous approaches.

2. Seven probing tasks, three probing methods, and different dialogue models provide a sufficient experiment setting for probing dialogue models.

3. The results are consistent with detailed and clear discussions.

**Reasons To Reject:**

1. The conclusions are not surprising. More in-depth probing experiments are expected, such as the analysis of understanding dialogue discourse relations, etc. The experiment setup for the impact of word embedding (using random word embeddings) and dialogue structure (shuffling the order of tokens) are too simple, leading to known conclusions.

2. The probing model may learn some shortcuts with the growing number of trainable parameters, which leads to performance gains.

**Reproducibility:**

4: Could mostly reproduce the results, but there may be some variation because of sample variance or minor variations in their interpretation of the protocol or method.

**Reviewer Confidence:**

4: Quite sure. I tried to check the important points carefully. It's unlikely, though conceivable, that I missed something that should affect my ratings.

**Typos Grammar Style And Presentation Improvements:**

Other suggestions:
The dialogue models are out-of-date to some extent. More recent models, such as vicuna-chat, alpaca, baize, etc.

[1] https://lmsys.org/blog/2023-03-30-vicuna/
[2] https://huggingface.co/project-baize/baize-v2-7b
[3] https://crfm.stanford.edu/2023/03/13/alpaca.html

---

> ### Author Rebuttal · Authors · 2023-08-29
>
> **Q1. Expect more in-depth probing experiments**
>
> Thank you for your feedback. While I appreciate the call for more in-depth probing experiments and your opinion of our experimental setup, I'd like to address your points:
>
> * Our experimental setup was designed to follow previous research in this area to maintain consistency and comparability. This allows for a more reliable assessment of how our work stands in relation to existing literature. Importantly, even within this established framework, our findings were not merely a reiteration of known conclusions but provided new and more credible insights.
>
> * We agree that additional probing experiments, such as the analysis of understanding dialogue discourse relations, would add value. However, it's worth noting that our current work already includes extensive experiments, the details of which consumed four pages in the appendix due to their complexity and depth. Given the constraints on article length, we found it prudent to leave further in-depth analyses for future work.
>
> ---
>
> **Q2. Suggestions of recent dialogue models (Vicuna, Alpaca, Baize, etc.)**
>
> The baseline dialogue models we chose follow prior work [7][8][9], with the aim of studying the natural language understanding capability of basic transformer-based dialogue models. In addition, we also compared the understanding capabilities of dialogue models pre-trained on dialogue corpora with different amounts of parameters and different architectures. For recent dialogue models such as Vicuna-chat, Alpaca, and Baize, it is difficult for us to conduct controllable and fair comparisons because their training process includes various instruction data and human preference-aligned training to enhance their semantic understanding capabilities, and there may be risks of data leakage during this process. We have also noticed that various benchmarks such as MMCU, Big-Bench, C-EVAL, and SuperCLUE, appeared to evaluate the capabilities of these models, which will be a focus of our future research.
>
> ---
>
> **Q3. Query about significance test comparison in Table 1**
>
> We compared the performances of word embeddings and encoder states under one specific probing method. If one of the two above input sources (under the same dataset, with the same probing method) passes the significance test, then it is super-scripted with an asterisk.
>
> The majority of encoder state results outperform word embedding results, indicating that encoder states contain more semantic features than word embeddings. This result, conversely, can corroborate the rationality of a particular probing methodology.
>
> ---
>
> **Q4. Correlation between probing method performance and fine-tuned parameters**
>
> Thank you for your insightful question regarding the relationship between the performance of the probing method and the number of fine-tuned parameters. To address this, we already conducted extension experiments where we scaled the original two-layer MLP to a parameter size comparable with that of the MSP. The detailed results of this experiment are available in the Appendix (see Table 5).
>
> Despite matching the parameter size of the MLP to that of the MSP, we observed that the MLP's performance remained sub-optimal. This underscores the idea that the architecture and operation mode of the MSP are intrinsically more effective than simply scaling up the number of parameters. In essence, the additional experiments validate that the effectiveness of the MSP method extends beyond mere parameter scaling, reinforcing the method's robustness in achieving its predefined objectives.
>
> ---
>
> **Q5. Why is the decoder fixed for PBP while fine-tuned for MSP?**
>
> In our study, the reason for using a fixed decoder in Prompt-based Probing while opting for a fine-tuned decoder in Multi-Source Probing (MSP) is rooted in the specific objectives and empirical context of each approach.
>
> Prompt-based Probing serves as a foundational generative probing method, leveraging the well-established practice of prompt learning. Consistent with much of the prior research in this field [3][4][5][6][7], we opted not to fine-tune the decoder. This is because retaining or removing a fixed decoder has been a standard procedure in previous generative probing methodologies, ensuring that our findings are comparable and validated against extant literature.
>
> In contrast, Multi-Source Probing (MSP) is an innovative probing approach that brings additional dimensions to the table. Specifically, MSP employs a fine-tunable decoder to handle tasks rich in a priori information more effectively. This approach allows us to capitalize on the latent understanding capabilities potentially encoded within the decoder parameters—something generally overlooked in traditional probing methods. By fine-tuning the decoder in MSP, we unlock the ability to re-utilize information that would otherwise be discarded, thereby providing a more comprehensive assessment of the language model's skill in grasping and modeling contextual cues.
>
> This differentiation in decoder treatment between the two methods serves to both align with existing research practices and introduce novel probing capabilities, fulfilling different aspects of our research objectives.
>
> ---
>
> **References**
>
> [3]https://aclanthology.org/P17-1080/
>
> [4]https://proceedings.neurips.cc/paper/2017/file/b069b3415151fa7217e870017374de7c-Paper.pdf
>
> [5]https://aclanthology.org/P18-1198/
>
> [6]https://groups.csail.mit.edu/sls/publications/2017/ICLR17_Belinkov.pdf
>
> [7]https://aclanthology.org/2020.nlp4convai-1.15/
>
> [8]https://arxiv.org/pdf/2104.09574.pdf
>
> [9]https://arxiv.org/pdf/2005.00719.

---

### Official Review · Reviewer_qTdP · 2023-08-15

**Typos Grammar Style And Presentation Improvements:** + line 998 Bart -> BARTl; line 999 Be…
**Soundness:** 3

**Excitement:**

2: Mediocre: This paper makes marginal contributions (vs non-contemporaneous work), so I would rather not see it in the conference.

**Missing References:**

N/A

**Paper Topic And Main Contributions:**

The paper proposes a Multi-Source Probing (MSP) method to probe the dialogue comprehension abilities of open-domain dialogue models. The method aggregates features from multiple sources (dialogue history, current utterance, integrated context) and conducts downstream tasks in a generative manner with the use of soft prompting that is consistent with the model pre-training process.

**Questions For The Authors:**

+ What is the parameter scale of the transformer trained from scratch?

**Reasons To Accept:**

+ The paper introduces an innovative Multi-Source Probing (MSP) technique, which is both original and pragmatic in nature.
+ Moreover, the paper presents enlightening empirical discoveries concerning the open-domain dialogue models' aptitude for comprehending dialogues.

**Reasons To Reject:**

+ The rationale behind the proposed multi-source attention mechanism remains ambiguous. Is it imperative to devise distinct attention mechanisms for various dialogue sources? Even in cases where pertinent information is distributed disparately within the dialogue history, could not a singular attention mechanism be capable of discerning the more significant segments? The necessity of employing a multi-source attention approach warrants further deliberation.
+ The freezing of decoder parameters in prompt-based probing, in contrast to the tunability in multi-source probing, could potentially lead to an unfair setting for prompt-based probing. Hence, I recommend that the authors conduct an additional experiment involving a variant of probing where decoder tunability is maintained.
+ The implementation details concerning the verbalizer are absent. How is the verbalizer designed?



**Reproducibility:**

3: Could reproduce the results with some difficulty. The settings of parameters are underspecified or subjectively determined; the training/evaluation data are not widely available.

**Reviewer Confidence:**

3: Pretty sure, but there's a chance I missed something. Although I have a good feel for this area in general, I did not carefully check the paper's details, e.g., the math, experimental design, or novelty.

---

> ### Author Rebuttal · Authors · 2023-08-29
>
> Thank you for your valuable comments!
>
> **Q1: Questioning the necessity and motivation of multi-source attention mechanism**
>
> In our study, we found that a singular attention mechanism often struggles to efficiently extract valuable information from the entire dialogue history while simultaneously giving appropriate weight to critical elements in the most recent interactions. This observation is corroborated by our ablation experiments (see Table 4), which demonstrate the limitations of using a singular attention mechanism for probing tasks in dialogue systems.
>
> In contrast, our multi-source attention mechanism is specifically designed to address these challenges. By incorporating distinct attention sub-components for different parts of the dialogue, we create a more focused and effective way of gleaning insights from both historical and current conversational data. This approach is not only intuitively aligned with the inherent structure of dialogues, but it is also concise and universally applicable across different dialogue architectures.
>
> Therefore, we contend that the multi-source attention mechanism is not merely an optional refinement but a necessary evolution for improving the probing capabilities of dialogue systems.
>
> ---
>
> **Q2: Concerns about fairness due to fixed decoder parameters in prompt-based probing**
>
> Thanks for the very valuable points, we have in fact included this part of the validation experiment in the ablation experiments section. We removed multi-source attention and late fusion from MSP, and this degenerate form is essentially a version of PBP that is consistent with MSP in terms of decoder tunability (see Ablation Experiment MSP w/o MS, Appendix A) .
>
> Prompt-based Probing is a generic probing method that utilizes prompt learning. Being the most primitive generative probing method, it is not fine-tuned in the decoder part, as fixing or removing the decoder is a common approach used in most of the previous works [3][4][5][6][7].
>
> One of the key innovations introduced by the MSP is the use of a fine-tunable decoder to carry out probing tasks that are richer in a priori information, while it utilizes the discarded information in a more efficient way. The probing methods commonly used in previous works lack the application of potential understanding capabilities that may be encoded in the decoder parameters. Thus, we introduce MSP to reapply the discarded information and better evaluate the language model's ability to understand and model contextual information.
>
> ---
>
> **Q3: Implementation details of verbalizer**
>
> Thank you for bringing attention to the absence of implementation details concerning the verbalizer. We appreciate the importance of this element for a comprehensive understanding of our approach. The verbalizer is a well-established method in prompt learning, and its implementation is based on existing literature [1][2]. Due to space constraints, we did not delve into the specific design of the verbalizer in our paper. However, we acknowledge the need for this information and will provide a more detailed description along with relevant citations in the final draft of the paper.
>
> ---
>
> **Q4: Parameter scale of the transformer**
>
> The parameter scale of the transformer is 37M (Please see Table 8)
>
> ---
>
> **References**
>
> [1]https://arxiv.org/pdf/2108.02035.pdf
>
> [2] https://aclanthology.org/2022.acl-demo.10.pdf
>
> [3]https://aclanthology.org/P17-1080/
>
> [4]https://proceedings.neurips.cc/paper/2017/file/b069b3415151fa7217e870017374de7c-Paper.pdf
>
> [5]https://aclanthology.org/P18-1198/
>
> [6]https://groups.csail.mit.edu/sls/publications/2017/ICLR17_Belinkov.pdf
>
> [7]https://aclanthology.org/2020.nlp4convai-1.15/

---

### Meta-Review · Area_Chair_RsJv · 2023-09-18

**Recommendation:** 4

**Metareview:**

This paper introduces a Multi-Source Probing (MSP) method to evaluate the dialogue comprehension abilities of open-domain dialogue models. The reviewers mostly agree on the potential significance and originality of the work, as it introduces an innovative method and presents enlightening empirical discoveries. However, there are some concerns and suggestions for improvement.

One reviewer questions the necessity of employing a multi-source attention mechanism and suggests exploring a variant of probing with decoder tunability. Another reviewer suggests more in-depth probing experiments and analyzing dialogue discourse relations. They also mention the possibility of shortcuts in the probing model due to the growing number of trainable parameters. Additionally, there are requests for clarification on the MSP and Prompt Based methods, the role of decoder models compared to encoder-decoder models, and the use of recurrent models. The implementation details of the verbalizer are also requested.

The reviewers generally find the paper sound, with sufficient support for its major claims, but some minor points need more support or details. The presentation and writing style are considered clear and easy to understand.

In terms of excitement, there is ambivalence among the reviewers. While the paper has merits like reporting state-of-the-art results and presenting a nice idea, there are also concerns about incremental work and the need for further revision.

Overall, the paper shows promise but should address the reviewers' concerns and suggestions to improve its clarity, originality, and significance.

---

### Decision · Program_Chairs · 2023-10-07

**Decision:**

Accept-Main

**Comment:**

This paper introduces a Multi-Source Probing (MSP) method to evaluate the dialogue comprehension abilities of open-domain dialogue models. The reviewers mostly agree on the potential significance and originality of the work, as it introduces an innovative method and presents enlightening empirical discoveries. However, there are some concerns and suggestions for improvement.

One reviewer questions the necessity of employing a multi-source attention mechanism and suggests exploring a variant of probing with decoder tunability. Another reviewer suggests more in-depth probing experiments and analyzing dialogue discourse relations. They also mention the possibility of shortcuts in the probing model due to the growing number of trainable parameters. Additionally, there are requests for clarification on the MSP and Prompt Based methods, the role of decoder models compared to encoder-decoder models, and the use of recurrent models. The implementation details of the verbalizer are also requested.

The reviewers generally find the paper sound, with sufficient support for its major claims, but some minor points need more support or details. The presentation and writing style are considered clear and easy to understand.

In terms of excitement, there is ambivalence among the reviewers. While the paper has merits like reporting state-of-the-art results and presenting a nice idea, there are also concerns about incremental work and the need for further revision.

Overall, the paper shows promise but should address the reviewers' concerns and suggestions to improve its clarity, originality, and significance.